# A health terminological system for inherited retinal diseases: Content coverage evaluation and a proposed classification

**Hamideh Sabbaghi[1,2], Sina Madani[3], Hamid Ahmadieh[4], Narsis Daftarian[5]\*, Fatemeh Suri[4], Farid Khorrami[6], Proshat Saviz[7], Mohammad Hasan Shahriari[8], Tahmineh Motevasseli[4], Sahba Fekri[4], Ramin Nourinia[4], Siamak Moradian[4], Abbas Sheikhtaheri** [7]\*

1 Ophthalmic Epidemiology Research Center, Research Institute for Ophthalmology and Vision Science, Shahid Beheshti University of Medical Sciences, Tehran, Iran, 2 Department of Optometry, School of Rehabilitation, Shahid Beheshti University of Medical Sciences, Tehran, Iran, 3 Department of HealthIT, Vanderbilt University Medical Center, Nashville, TN, United States of America, 4 Ophthalmic Research Center, Research Institute for Ophthalmology and Vision Science, Shahid Beheshti University of Medical Sciences, Tehran, Iran, 5 Ocular Tissue Engineering Research Center, Research Institute for Ophthalmology and Vision Science, Shahid Beheshti University of Medical Sciences, Tehran, Iran, 6 Department of Health Information Technology, Hormozgan University of Medical Sciences, Bandar Abbas, Iran, 7 Department of Health Information Management, School of Health Management and Information Sciences, Iran University of Medical Sciences, Tehran, Iran, 8 Department of Health Information Technology and Management, School of Allied Medical Sciences, Shahid Beheshti University of Medical Sciences, Tehran, Iran

\* sheikhtaheri.a@iums.ac.ir (AS); nardaftarian@hotmail.com (ND)

**Data Availability Statement:** Additional data has been uploaded to the journal website as Supporting Information files.

## Abstract

### Purpose

To present a classification of inherited retinal diseases (IRDs) and evaluate its content coverage in comparison with common standard terminology systems.

### Methods

In this comparative cross-sectional study, a panel of subject matter experts annotated a list of IRDs based on a comprehensive review of the literature. Then, they leveraged clinical terminologies from various reference sets including Unified Medical Language System (UMLS), Online Mendelian Inheritance in Man (OMIM), International Classification of Diseases (ICD-11), Systematized Nomenclature of Medicine (SNOMED-CT) and Orphanet Rare Disease Ontology (ORDO).

### Results

Initially, we generated a hierarchical classification of 62 IRD diagnosis concepts in six categories. Subsequently, the classification was extended to 164 IRD diagnoses after adding concepts from various standard terminologies. Finally, 158 concepts were selected to be classified into six categories and genetic subtypes of 412 cases were added to the related concepts. UMLS has the greatest content coverage of 90.51% followed respectively by SNOMED-CT (83.54%), ORDO (81.01%), OMIM (60.76%), and ICD-11 (60.13%). There

**Funding:** The authors received no specific funding for this work.

**Competing interests:** The authors have declared that no competing interests exist.

were 53 IRD concepts (33.54%) that were covered by all five investigated systems. However, 2.53% of the IRD concepts in our classification were not covered by any of the standard terminologies.

## Conclusions

This comprehensive classification system was established to organize IRD diseases based on phenotypic and genotypic specifications. It could potentially be used for IRD clinical documentation purposes and could also be considered a preliminary step forward to developing a more robust standard ontology for IRDs or updating available standard terminologies. In comparison, the greatest content coverage of our proposed classification was related to the UMLS Metathesaurus.

## Introduction

Inherited retinal diseases (IRDs) consist of various types of hereditary retinal dystrophies that are primarily involving the outer retina and retinal pigment epithelium (RPE) [1, 2]. Considering the progressive nature of IRDs which can lead to decreased vision-related quality of life (VRQoL), accurate diagnosis and identification of the causative gene mutations are important for genetic consulting as well as allocating research-based trials for the evolving therapies such as gene therapy [3–5]. Therefore, domain-specific ontologies and reference terminology systems are considered important tools that can facilitate documentation and representation of the complex IRD concepts based on their genotype-phenotype distinctions [6–8].

Clinical classifications and terminology systems are widely used for standardizing patients' health information at the point of care for different purposes such as clinical, research, reimbursement, and epidemiological purposes [9–11]. Additionally, terminological systems are essential for structured documentation of patient data in Electronic Health Record (EHR) systems [12]. One of the crucial advantages of these systems is the establishment of a common controlled vocabulary for effective communication by capturing unified medical terms and concepts [13]. Furthermore, such structured contents can be reused for a variety of purposes and represented in many different information systems [10–15].

Based on the intended application, different terminological systems are available in the domain of medicine including Unified Medical Language System (UMLS), Online Mendelian Inheritance in Man (OMIM), International Classification of Diseases (e.g., ICD-11), Systematized Nomenclature of Medicine—Clinical Terms (SNOMED-CT) and Orphanet Rare Disease Ontology (ORDO) [16]. Each system has been developed for a particular purpose. For instance, UMLS is widely used for providing a unique code and term linkage between different terminological systems and facilitating mapping schemas that can be used to create and maintain local terminologies [17]. While ICD classifications can be employed for the identification of health trends, mortality and morbidity statistics [18] as well as clinical and health research purposes; OMIM is optimized for the classification of human genes and genetic disorders [19]. Recently, SNOMED-CT has been designated as the terminology of choice for the standardization of complex medical concepts, such as IRDs, at the point of care [20]. Additionally, Orphanet also represents a dedicated coding system for rare diseases by a stable and unique ORPHA number [8]. Since the above terminologies cover the IRDs domain, we have incorporated them all throughout our study.

An ontology consists of concepts that organize domain knowledge with defined terms, concepts, and their relationships in a given field of study. Ontologies are useful both for providing consistent domain concepts and representing structured data formats in order to facilitate the knowledge discovery process [21]. With regards to the importance of terminological systems in the recorded history of biological and natural sciences [22], Sergouniotis et al. [8] presented a hierarchical ontology for ocular phenotypes and rare ocular diseases that covered all the rare diseases not only IRD concepts. In addition, only the entities were extracted based on the two systems of Human Phenotype Ontology and Orphanet Rare Disease Ontology and did not include the genetic concepts. While in the present study, we desired to develop a dedicated classification, specifically for IRD concepts.

One of the preliminary steps in selecting or developing a new classification or ontology system is to search for existing ones. Evaluation of the content coverage of existing (or new) ontologies is a quintessential feature of each terminological (ontological) system. Content coverage presents the extent of content representation (either term or concept) for a developed terminology, based on the standards and is highly recommended to be evaluated in order to expand the adequacy of the content representation [10, 23]. A systematic review shows that there are few studies focusing on the content coverage evaluation of terminological systems in the ophthalmology domain [10]. In the study by Hwang et al. [12], content coverage analysis was conducted on five controlled terminological systems of ICD-9, Logical Observation Identifiers, Names, and Codes (LOINC), SNOMED-CT, Medical Entities Dictionary (MED), and Current Procedural Terminology, fourth edition (CPT-4). It was concluded that the content coverage of these five evaluated systems was not perfect for general ophthalmic concepts. In another study by Chiang et al. [13], higher content coverage was found for the SNOMED-CT system for general ophthalmic concepts in comparison with ICD-9, CPT-4 and MED terminological systems.

Lack of consistent coding of IRD concepts by terminological systems may result in documenting and extracting unreliable information from electronic systems including patient registries and electronic health records (EHRs). These issues may raise the challenges of semantic interoperability among electronic systems [24]. The objective of this study was two-sided: Firstly, we developed a classification for IRD concepts, and secondly, we evaluated the content coverage of some well-known terminological systems for our proposed IRD classification to validate this classification for documentation of IRD concepts.

## Results

### Our proposed classification for IRD concepts

Initially, during our literature review, we created a prototype classification that contained 62 IRD-related concepts based on anatomical and functional features as well as genetic characteristics in six categories (S1 Table). Gradually, such classification was expanded to 164 IRD concepts under the six main categories, represented by standard terminologies. During expert consultation sessions, several panel group discussions, and further literature review, the consensus agreed to retract eleven concepts from the proposed classification because they were not fully representing an IRD (S2 Table). Therefore, a final list was developed that contained 158 IRD concepts in six categories based on the phenotype identifications and their corresponding genetic subtypes (n = 412). In our proposed classification, the disease categorization was organized considering both anatomical and functional IRD entities in the following six categories: 1. Diffuse photoreceptor dystrophies (n = 15), 2. Macular dystrophies (n = 22), 3. Chorioretinal dystrophies (n = 12), 4. Inner retinal and/or vitreoretinal dystrophies (n = 4), 5. Retinal dystrophies associated with systemic disease (syndromic diseases) (n = 92), and 6.

Congenital & stationary retinal diseases (n = 13). For further clarification of the twenty- one IRD concepts, we classified them under our proposed categories, which are performed based on the updated anatomic, functional and genetic information, although might be different from the previous classification of the standard terminological systems (S3 Table). Our final IRD classification is registered on the BioPortal website and can be accessed via this web address https://bioportal.bioontology.org/ontologies/IRD3.

## Content coverage (term and concept) analysis

We evaluated our IRD classification against 5 terminological systems that we included in our scope. In general, UMLS had the greatest coverage of 90.51% compared with other evaluated systems including SNOMED-CT (83.54%), ORDO (81.01%), OMIM (60.76%), and ICD-11 (60.13%). Overall, there were 53 IRD concepts (33.54%) that were covered by all five investigated systems. However, 2.53% of the IRD concepts in our classification were not covered by any of the standard terminologies (S4 Table; for genetic classification, see Supplementary Excel sheet labeled as Terminological System for IRDs). Table 1 summarizes the coverage analysis of the developed IRD classification based on SNOMED-CT. Our classification was completely covered based on both terminologies and concepts within SNOMED-CT regarding diseases for chorioretinal dystrophies, and inner retinal and/ or vitreoretinal dystrophies. Even though there was no match for 16.46% (n = 26) of IRD concepts in SNOMED-CT, we were able to create an equivalent for 6.96% (n = 11) of the concepts by leveraging the post-coordination function in SNOMED-CT.

Table 2 shows content coverage for the remaining 4 terminologies (ICD-11, ORDO, OMIM, and UMLS), of our suggested IRD classification. UMLS and SNOMED-CT had the highest content coverage for all IRD categories, while the least content coverage was reported for ICD-11.

## Discussion

This study proposes a classification for IRD concepts and terminologies. This will benefit subspecialists in the field of IRD diseases, who are trying to go to a nomenclature system that facilitates the allocation of disorders into precise categories based on individual gene mutations, which is the result of improving diagnostic techniques. Evaluation of content coverage was based on the UMLS, ICD- 11, ORDO, OMIM and SNOMED- CT terminologies. The highest content coverage was obtained for the UMLS system due to the generalizability of this system for a variety of biomedical vocabularies [17].

One of the shortcomings of OMIM is that its concept coding is not based on a uniform pattern, hence, various IRD concepts with the same underlying gene were considered as a distinct diagnosis, while in some other cases one concept was defined as the main one and the other one was considered as an alternative term. For instance, RP14 "OMIM: # 600132" and LCA15 "OMIM: # 613843" were considered as two distinct phenotypes, although TULP1 is a single causative gene of both diagnoses. Conversely, LCA13 "OMIM: # 612712" is considered as the main term and RP53 "OMIM: # 612712" is defined as its alternative term when RDH12 is the causative gene for both phenotypes. To resolve this discrepancy, we considered a unique code for each causative gene, despite having various phenotypes, in our proposed classification. In addition, we adopted this approach in cases when a single gene was the causing factor of various IRD diagnoses. Another limitation of the OMIM terminology system is that all subtypes of each IRD concept were determined only by a single OMIM code like "Retinitis punctata albescens; # 136880", whereas, in our proposed classification, each subtype has a unique code.

**Table 1. Coverage analysis of the IRD ontology based on SNOMED-CT terminological system.**

| IRD Concepts Categories | Coverage Analysis | | | | |
|---|---|---|---|---|---|
| | Term | | Concept | | |
| | No- matched | Matched | No- matched | Matched | Partially matched* |
| | n (%) | n (%) | n (%) | n (%) | n (%) |
| Diffuse Photoreceptor Dystrophies | 4 (26.67) | 11 (73.33) | 4 (26.67) | 10 (66.67) | 1 (6.66) |
| Macular Dystrophies | 4 (18.18) | 18 (81.82) | 4 (18.18) | 17 (77.27) | 1 (4.55) |
| Chorioretinal Dystrophy | 0 (0.0) | 12 (100) | 0 (0.0) | 9 (75.0) | 3 (25.0) |
| Inner Retinal and/or Vitreoretinal Dystrophies | 0 (0.0) | 4 (100) | 0 (0.0) | 4 (100) | 0 (0) |
| Retinal Dystrophies Associated with Systemic Disease (Syndromic Diseases) | 16 (17.39) | 76 (82.61) | 16 (17.39) | 72 (78.26) | 4 (4.35) |
| Congenital & Stationary Retinal Diseases | 2 (15.38) | 11 (84.62) | 2 (15.38) | 9 (69.24) | 2 (15.38) |

IRD, inherited retinal dystrophy; SNOMED-CT, Systematized Nomenclature of Medicine—Clinical Terms; n, number

* Partially matched were considered as same as post coordinated terms in cases with no possibility of term matching and covering by alternative terms.

**Table 2. Coverage analysis of the IRD ontology based on the terminological systems.**

| IRD Concepts Categories | Terminological Systems | Coverage Analysis | |
|---|---|---|---|
| | | No-matched | Matched |
| | | n (%) | n (%) |
| Diffuse Photoreceptor Dystrophies | ICD-11 | 6 (40.0) | 9 (60.0) |
| | ORDO | 5 (33.33) | 10 (66.67) |
| | OMIM | 5 (33.33) | 10 (66.67) |
| | UMLS | 1 (6.67) | 14 (93.33) |
| Macular Dystrophies | ICD-11 | 9 (40.90) | 13 (59.10) |
| | ORDO | 4 (18.18) | 18 (81.82) |
| | OMIM | 11 (50.0) | 11 (50.0) |
| | UMLS | 3 (13.64) | 19 (86.36) |
| Chorioretinal Dystrophies | ICD-11 | 2 (16.67) | 10 (83.33) |
| | ORDO | 3 (25.0) | 9 (75.0) |
| | OMIM | 5 (41.67) | 7 (58.33) |
| | UMLS | 2 (20.0) | 10 (80.0) |
| Inner Retinal and/or Vitreoretinal Dystrophies | ICD-11 | 1 (25.0) | 3 (75.0) |
| | ORDO | 0 (0) | 4 (100) |
| | OMIM | 1 (25.0) | 3 (75.0) |
| | UMLS | 0 (0) | 4 (100) |
| Retinal Dystrophies Associated with Systemic Disease (Syndromic Diseases) | ICD-11 | 41 (44.56) | 51 (55.44) |
| | ORDO | 15 (16.30) | 77 (83.70) |
| | OMIM | 32 (34.78) | 60 (65.22) |
| | UMLS | 6 (6.52) | 86 (93.48) |
| Congenital & Stationary Retinal Diseases | ICD-11 | 4 (30.77) | 9 (69.23) |
| | ORDO | 3 (23.07) | 10 (76.93) |
| | OMIM | 8 (61.54) | 5 (38.46) |
| | UMLS | 3 (23.07) | 10 (76.93) |

IRD, Inherited Retinal Dystrophy; ICD, International Classification of Diseases; ORDO, Orphanet Rare Disease Ontology; OMIM, Online Mendelian Inheritance in Man; UMLS, Unified Medical Language System; n, number

Acceptable content coverage was observed in ICD-11 for non-genetic based IRD concepts; however, its accuracy needs to be further evaluated. For instance, the same code of 5C56.00 is assigned to different types of GM Ganliosidosis.

Although a high level of content coverage was obtained for UMLS Metathesaurus, it is not widely incorporated in electronic health record and data entry systems. Therefore, enhancing content coverage of the more commonly used clinical terminologies such as ICD and SNO-MED-CT could facilitate a wider adoption.

There were eleven IRD diagnoses that are represented ORDO classification system; however, we decided to exclude them from our proposed classification. IRD concepts that were removed from our classification, together with a brief description of the reason, are as follow:

a. *Albers-Schönberg Osteopetrosis*
This entity is classified as genetic macular dystrophy in ORDO system [25], while blindness, optic atrophy and visual loss are only the main ophthalmic complications reported by different studies cited by OMIM system [26–29]. This concept is categorized as a type of "hereditary disorder of the musculoskeletal system" by SNOMED-CT and ICD-11 considered it as a subtype of osteopetrosis diagnosis [30].

b. *Foveal Hypoplasia—Presenile Cataract Syndrome*
This concept is classified as IRD by ORDO without additional explanation [31]. UMLS provides a CUI with no explanation, while it is not represented in SNOMED-CT. It is classified as a subtype of "other cataract" related to "Infantile and juvenile cataract" classification. Based on OMIM results, it is an ocular disease which is presented with aniridia.

c. *Hermansky—Pudlak Syndrome with Neutropenia*
SNOMED-CT does not consider this concept as IRD and classifies it as a metabolic disorder by ICD- 11 as well [32]. A CUI code is assigned for this concept by UMLS system with no additional explanation. In ORDO and OMIM, oculocutaneous albinism is a known ocular finding in these patients [33].

d. *Leigh Syndrome*
This disease is manifested by involvement of the central nervous system (CNS) resulting in ophthalmoplegia as a common ocular finding. OMIM considers it as a disorder of the CNS with no relation to any specific ocular disease [34]. Vision loss and eye movement abnormality are the common ocular findings, as shown in UMLS. Both SNOMED-CT and ICD-11 considered this diagnosis as a hereditary disorder of the CNS, as well [35].

e. *Metachromatic Leukodystrophy*
ICD- 11 system considered it as a subtype of metabolic disease [36], but UMLS and SNO-MED-CT consider it as a hereditary disorder of the nervous system with no ocular manifestation. OMIM system reports no ocular finding [37].

f. Rubinstein—Taybi Syndrome
It is considered as congenital glaucoma by OMIM system [38]. Glaucoma, refractive error and nasolacrimal duct obstruction are common ocular findings reported by ORDO system [39]. SNOMED-CT classifies this concept as hereditary cancer predisposing syndrome. No ocular findings were reported by UMLS system. ICD- 11 also considers it as a congenital anomaly [40].

g. *Retinal Vasculopathy with Cerebral Leukoencephalopathy and Systemic Manifestations*
SNOMED-CT does not specify any code for this diagnosis, while UMLS has a CUI for it with no definition. Hereditary vascular retinopathy, hereditary endotheliopathy with

retinopathy as well as progressive visual impairment are all represented as ocular manifestations in ORDO [41].

h. *Revesz syndrome*

This concept is classified as IRD by ORDO with ocular manifestation of bilateral exudative retinopathy [42]. The same definition exists in UMLS. This concept is also covered by OMIM, without any relationship to retinal dystrophies [43]. This concept is classified as a subtype of hereditary disorder of the visual system by SNOMED-CT.

i. *Aceruloplasminemia*

This clinical diagnosis is classified as IRD by UMLS, while it is considered a metabolic disease by SNOMED-CT.

j. *Cockayne Syndrome*

Photosensitivity is only the visual finding which is included in ORDO [44]. Additionally, the common ocular manifestations represented in UMLS are photosensitivity, vision impairment, and retinal pigmentation. Congenital cataract is a single ocular finding reported by OMIM, while ICD- 11 emphasizes on congenital malformation in the affected individuals [45]. SNOMED-CT considers it a subtype of hereditary disorder of the nervous system.

k. *Xeroderma Pigmentosum—Cockayn Syndrome Complex*

There is no concept code for this diagnosis in UMLS or SNOMED-CT. ORDO classifies it as IRD [46], while ICD- 11 only emphasizes on congenital malformation [47]. There is no finding suggestive of retinal dystrophy presented by OMIM.

Although the proposed classification is the first dedicated hierarchy for subtypes of IRD diagnoses, a suitable linkage to standard terminologies is highly recommended which would be of importance for standardization, maintenance, and reusability of this classification.

The primary purpose of the present study is to present a classification for IRDs and we intend to continue our study focusing on the comparison of our classification system with concepts identified in literature by Natural Language Processing (NLP) algorithms and implementing of this classification in the form of an interface terminology and evaluating user perspectives in terms of usability and clinical applications. Additionally, generation of a mapping between IRD concepts and the target terminologies was a side benefit of our study. Considering the fact that such IRD concepts are inherited via genetic mutations, coding IRD concepts with regards to their genetic mutations is highly desirable.

Although there is no definite cure for IRDs at present, accurate diagnosis and identification of the causative gene mutations can facilitate the creation of animal models for providing gene therapy for IRDs. Therefore, any classification system like ontologies or reference terminologies would become essential tools for personalized medicine specific to IRD diseases and facilitate the description and specification of complex IRD concepts based on the genotype-phenotype distinction.

Considering functional and anatomical attributes, within seven categories, in our IRD classification is the most prominent factor in the proposed classification, which would certainly be helpful in designing future studies focusing on each IRD subtype.

Considering the fact that the World Health Organization (WHO) has already developed the ICD-11 Ophthalmology Specialty Linearization, the proposed classification can be considered a hierarchical prototype for the particular type of ophthalmic diseases.

In conclusion, the specialized classification system that we created, including terms and concepts, is dedicated to various types of IRD diagnosis and has the potential to be applied in

the clinics and electronic information systems. More importantly, it may be considered as a preliminary step toward development of a standard ontology for inherited retinal diseases. In addition, the study indicates that many of available terminology systems has little coverage of IRD concepts and only UMLS covers more than 90 percent of concepts. Therefore, our classification system can also be applied to develop and update further versions of standard classification systems.

## Methods

In our proposed classification system, each IRD disease is mainly grouped based on the main origin of its pathological problem considering both anatomical and functional concepts. Three steps were taken in our study to develop IRD classification and evaluate the content coverage of terminological systems regarding the IRD concepts.

### Step I: Development of a classification for IRD concepts

Initially, a working group of two academic, board-certified retina specialists, one academic molecular geneticist, and five medical informatics specialists was organized. Then, a comprehensive literature review on IRDs was conducted including the textbooks [48–50], and original and review articles [5, 51–80] in order to develop a primary classification based on the anatomical, functional, and pathological concepts related to IRDs. Afterwards, a comprehensive search was conducted up to December 2022 on five terminological systems, including UMLS, ICD-11, SNOMED-CT, ORDO and OMIM to look for any existing terminology for different genetic and phenotypic IRDs to be added accordingly to our primary classification. Then, a questionnaire was developed based on the identified diagnoses, where it was discussed with an expert panel of nine academic board-certified retina specialists regarding the scientific and conceptual accuracy of each IRD concept. Ten expert panel discussion sessions with an interval of one month were held in order to collect all comments and feedback. In each session, 40 IRD concepts were reviewed. After this phase, some irrelevant or duplicated IRD concepts were dropped from the classification list and some concepts were classified in the updated categories based on their functional and anatomical characteristics. All participants were asked to provide "yes", "no" or "maybe" response to "validated", "not validated", and "suspected" concepts, respectively. IRD concepts that were labelled as "maybe" were discussed in the subsequent sessions of the expert panel to yield confirmation. This process continued until reaching a final decision. The concepts labeled as "yes" with an agreement of 70% or more were considered as IRD concept, otherwise they were defined as "not validated" IRD concept. To obtain a comprehensive consensus, the summary of the results of the previous sessions was presented in the succeeding meetings. Eventually, after applying the proposed changes to the classification it was finalized based on the consensus of the panel group members based on the workflow represented in Fig 1. Finally, an organized hierarchy of IRD concepts was developed in an ontology editor platform called Protégé (Fig 2). This proposed classification was structured based on the phenotype characteristics and the genetic concepts were included to the related IRD diagnosis.

### Step II: Extraction of annotation for each IRD concept

In this phase, all annotations for each IRD concept including code, fully specified name, synonyms (alternative descriptions) in other terminological systems were extracted by two experts: a vision scientist and a medical informatics specialist. This information was documented in a separate datasheet for further comparison. We also assigned a unique internal code for each IRD concept in order to create a mapping network for future data integration.

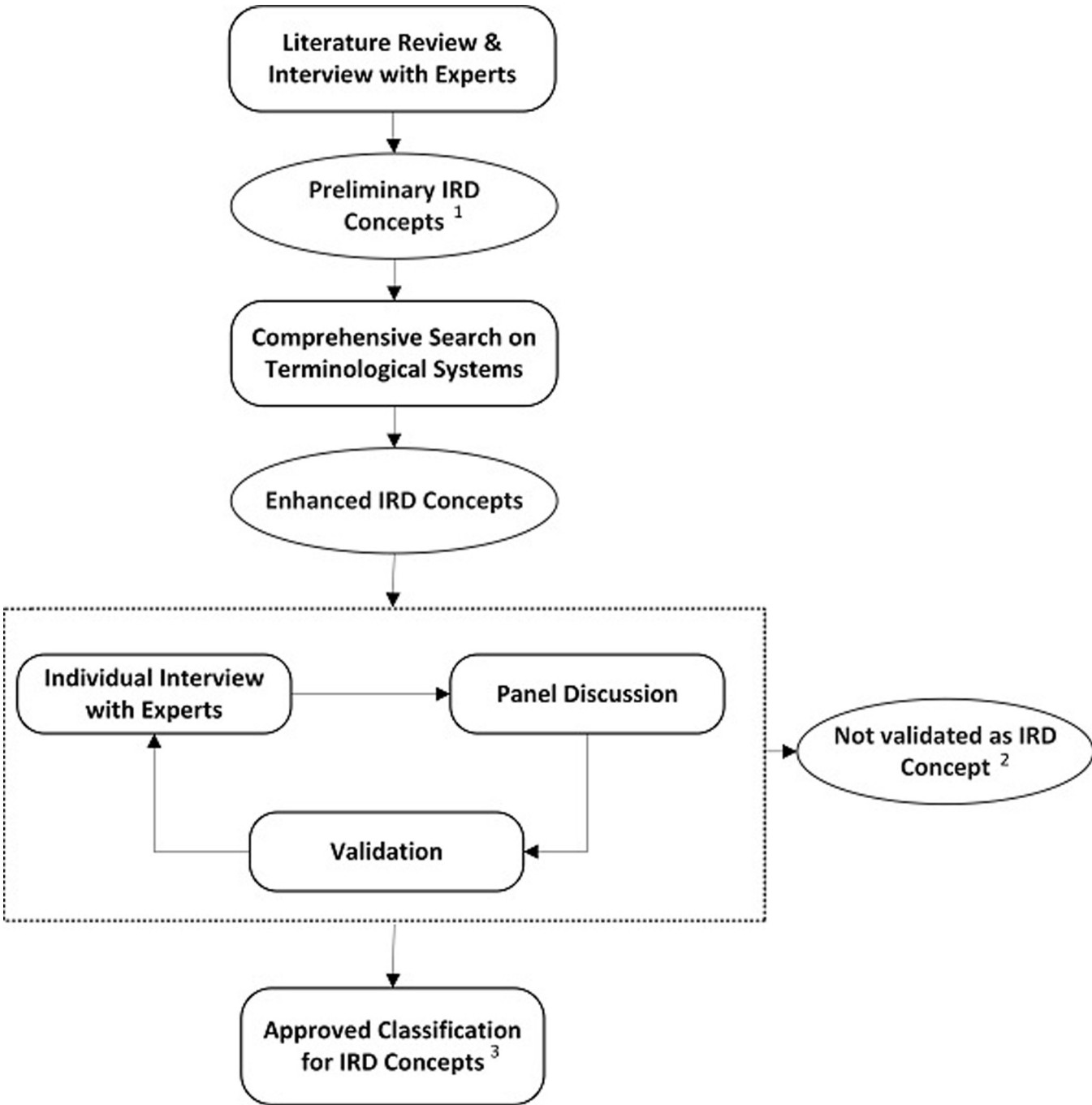

**Fig 1. Our workflow to develop the proposed classification.** IRD, inherited retinal dystrophy. [1] Presented on Supplementary data, Table 1. [2] Presented on Supplementary data, Table 2. [3] Developed by Protégé software and it could be accessible on BioPortal website through the URL address of http://bioportal. bioontology.org/ontologies/IRD3.

## Step III: Comparison among terminological systems for content coverage analysis

Content coverage analysis was conducted from two aspects; terms and concepts [81]. As previously stated, term- and concept- coverage are considered as the extent of term and concept

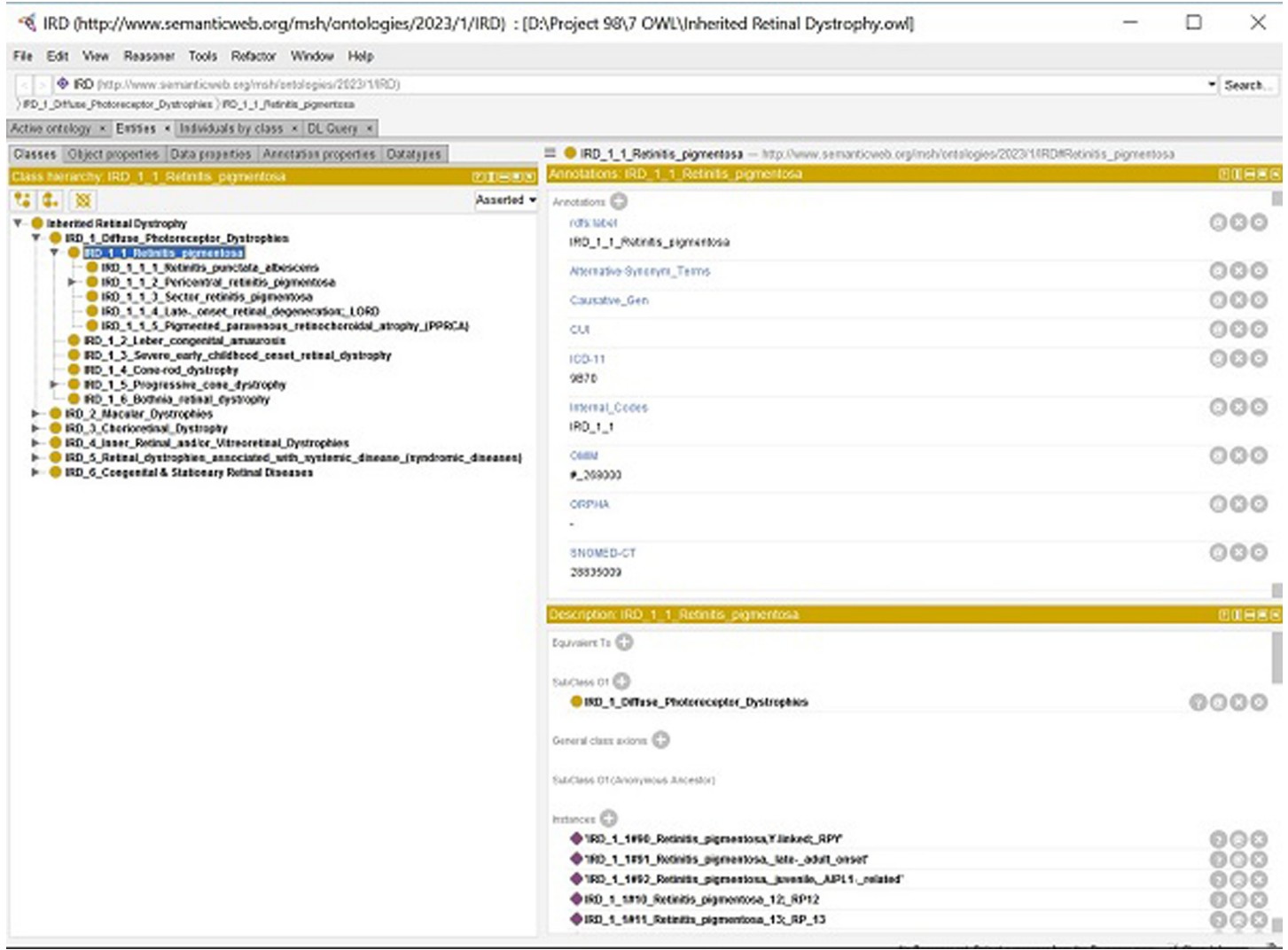

**Fig 2. A screenshot of registered inherited retinal dystrophy concepts in Protégé software.**

representation of our proposed classification based on the five mentioned terminological systems.

Initially, each IRD concept in our classification was checked against terminological systems in order to find the relevant or equivalent terms and/or concepts. We classified our results as "no match", "partial match" or "complete match". An IRD concept was considered as term- and concept- matched if the term was completely representable throughout the above-mentioned terminological systems. Further evaluation regarding concept matching was performed for no-term-matched IRD contents. "Complete concept match" was defined in cases where alternative terms (or synonymous) existed. In SNOMED-CT, we considered post-coordination if no synonym was found. In fact, if it was possible to develop a concept with post-coordination, we considered that concept as "partial match". IRD contents with no possibility of post-coordination were considered as "complete no match". The percentage for each of the above- mentioned conditions was reported by the terminological systems. Our content

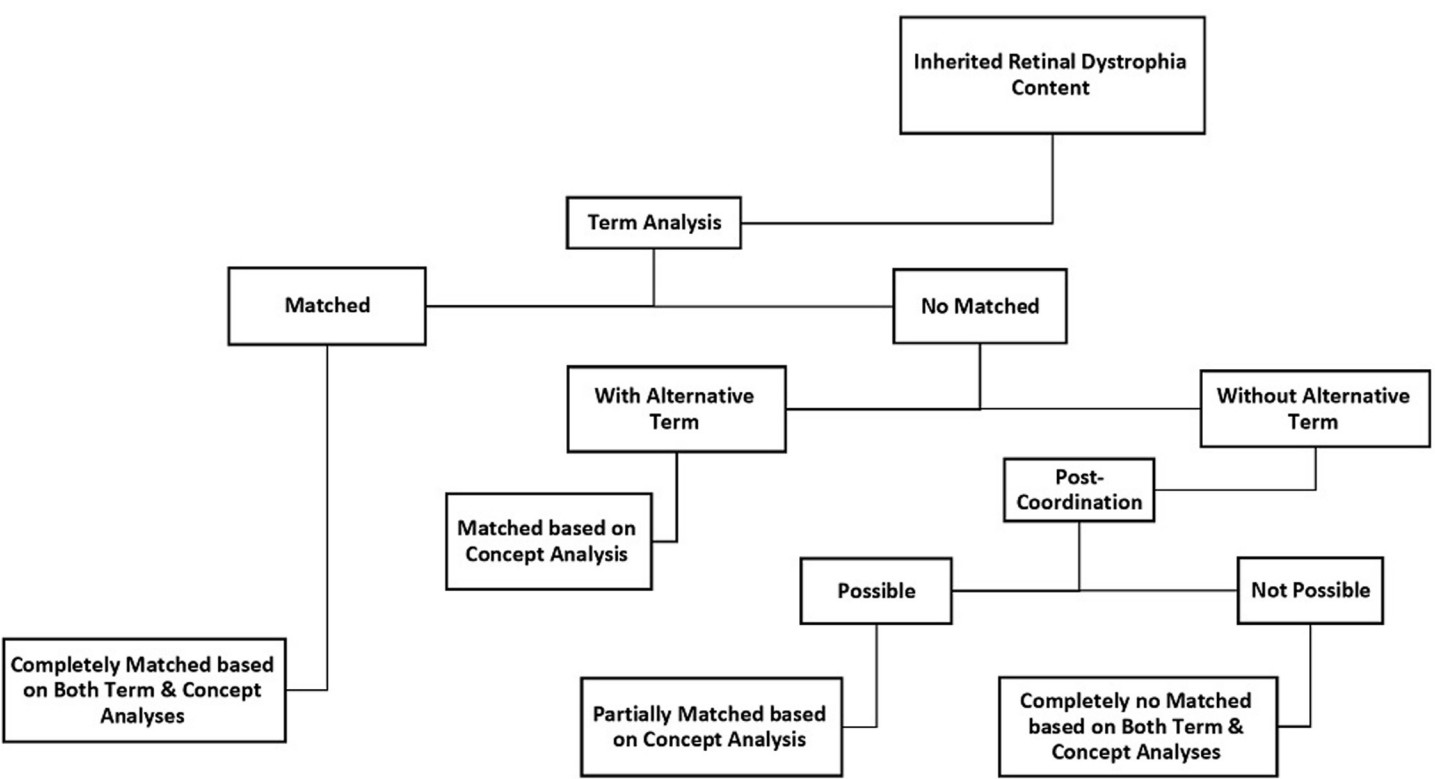

**Fig 3. Our workflow to evaluate the content coverage of the proposed classification based on the standard terminological systems.**

coverage approach is illustrated in Fig 3. An example of the content coverage analysis of IRD concepts based on the SNOMED- CT system is shown in Fig 4.

## Declarations

### Ethics approval and consent to participate

The study details were presented to the Ethics Committee of the Ophthalmic Research Center affiliated to Shahid Beheshti University of Medical Sciences and approved via the registration number of IR.SBMU.ORC.REC.1396.15. This article does not contain any studies with human participants or animals performed by any of the authors, therefore the informed consent was waived based on the local Research Ethics Committee decision.

| IRD Concept | Term Matched | Matched based on Concept Analysis (Alternative Term) | Partially Matched (Post- coordination) | No- matched |
|---|---|---|---|---|
| Retinitis pigmentosa | ✓ | - | - | - |
| Adult-onset foveomacular vitelliform dystrophy | × | ✓ (adult vitelliform macular dystrophy) | – | – |
| Pericentral retinitis pigmentosa | × | × | ✓ (pericentral + retinitis pigmentosa) | – |
| Dominant cystoid macular dystrophy | × | × | × | ✓ |

**Fig 4. Representation of examples for different levels of term- and concept- matching for inherited retinal dystrophy (IRD) concepts based on SNOMED-CT.**

## Supporting information

**S1 File. All terminological systems and causative gene.**
(XLSX)

**S1 Table. A preliminary classification of inherited retinal dystrophy (IRD) concepts.**
(DOCX)

**S2 Table. Not validated concepts as inherited retinal dystrophy (IRD).**
(DOCX)

**S3 Table. Classification of IRD concepts in our proposed ontology and applied standard terminological system.**
(DOCX)

**S4 Table. A list of IRD concepts coded by all systems and those coded by none of systems.**
(DOCX)

## Acknowledgments

The authors desire to express their deep gratitude to the research subcommittee of the Iranian National Registry for Inherited Retinal Diseases (IRDReg®).

## Author Contributions

**Conceptualization:** Sina Madani, Abbas Sheikhtaheri.

**Data curation:** Fatemeh Suri.

**Investigation:** Abbas Sheikhtaheri.

**Methodology:** Hamideh Sabbaghi, Sina Madani, Proshat Saviz, Abbas Sheikhtaheri.

**Project administration:** Hamideh Sabbaghi, Hamid Ahmadieh, Narsis Daftarian, Farid Khorrami, Proshat Saviz, Abbas Sheikhtaheri.

**Software:** Farid Khorrami, Mohammad Hasan Shahriari.

**Supervision:** Hamideh Sabbaghi, Hamid Ahmadieh, Farid Khorrami, Abbas Sheikhtaheri.

**Validation:** Abbas Sheikhtaheri.

**Writing – original draft:** Hamideh Sabbaghi, Hamid Ahmadieh, Narsis Daftarian, Fatemeh Suri, Proshat Saviz, Abbas Sheikhtaheri.

**Writing – review & editing:** Hamideh Sabbaghi, Hamid Ahmadieh, Narsis Daftarian, Fatemeh Suri, Farid Khorrami, Tahmineh Motevasseli, Sahba Fekri, Ramin Nourinia, Siamak Moradian, Abbas Sheikhtaheri.

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
