## [Decision Letter · Decision Letter 0]

18 Jul 2022

PONE-D-22-12220A Health Terminological System for Inherited Retinal Diseases: Content Coverage Evaluation and a Proposed Novel ClassificationPLOS ONE

Dear Dr. sheikhtaheri,

Thank you for submitting your manuscript to PLOS ONE.  After careful consideration, we feel that it has merit but does not fully meet PLOS ONE’s publication criteria as it currently stands. Therefore, we invite you to submit a revised version of the manuscript that addresses the points raised during the review process. I am sorry that it has taken a long time to present you with a review, but we were trying to find a second expert to review your submission.  In the interests of time, I decided to ask for revision based on the comments of one expert whose comments are valuable.

Please modify your comments about the novelty of your work, because there are already several classification systems for IRDs, though none is widely accepted. Please explain how genetic data informed your classification scheme.  The reviewer indicates that including genotypic information is important.

We look forward to receiving your revised manuscript.

Kind regards,

Alfred S Lewin, Ph.D.

Section Editor

PLOS ONE

Journal Requirements:

Reviewers' comments:

Reviewer's Responses to Questions

**Comments to the Author**

1. Is the manuscript technically sound, and do the data support the conclusions?

Reviewer #1: Partly

2. Has the statistical analysis been performed appropriately and rigorously? 

Reviewer #1: N/A

3. Have the authors made all data underlying the findings in their manuscript fully available?

Reviewer #1: Yes

4. Is the manuscript presented in an intelligible fashion and written in standard English?

Reviewer #1: Yes

5. Review Comments to the Author

Reviewer #1: The authors present a novel IRD classification. The methodology used to achieve this new classification is well defined and all data was made available. Please find my queries below:

Abstract

– Purpose: Please use “To present a novel…” instead of “To represent a novel…”

Introduction

– The authors write: “to the best of our knowledge there is no dedicated ontology or classification for IRD concepts”. This is not true. There are several proposed classifications. However, I would say there is not one widely accepted classification. Also, please check PMID: 30626441 (ref 8 in your references list). The authors present a summary of the modifications made to HPO and ORDO as part of the study (Table S1), and list the disease groups and specific disorders introduced to ORDO as a result of the ERN-EYE Ontology meeting (Mont Sainte-Odile, France, 10/2017). The ERN-EYE is an European Reference Network covering all rare eye disorders and not only IRDs. The group has worked hand in hand with ORPHANET to improve IRD and other rare eye disease nomenclature in ORDO.

– Please replace “…to validate the value of this novel…” for “…to validate this novel…”

Methods

– Step I: the authors write “This proposed classification was structured based on the phenotype characteristics and the genetic concepts were included to the related IRD diagnosis.” Please elaborate on the genetic concepts. I cannot find in the text, tables, figures or supplemental material any reference to the genetic cause of the proposed diagnoses. In fact, I would say the major issue with the presented classification is that it lacks genetic information. Please elaborate on this as it is a fundamental part of any new IRD classification. Several genes are able to cause different phenotypes and gene-based classifications are getting more popular. Developing a strong phenotypical classification is important but it must incorporate genetic data, ie, genes associated with each phenotype and must allow seeing things the other way around, ie which phenotypes are associated with each gene.

– Another issue is that some diagnoses fit more than one category. An example would be central areolar choroidal dystrophy (CACD) which fits the macular dystrophies group and choroidal dystrophies group. The same is true for syndromic forms of retinitis pigmentosa (eg Usher, BBS) that fit the Retinal dystrophies associated with systemic disease (syndromic diseases) group and the Diffuse Photoreceptor Dystrophies group; or syndromic forms of cone-rod dystrophy (eg. Cone-rod dystrophy and hearing loss, caused by CEP250 mutations) that fit the Retinal dystrophies associated with systemic disease (syndromic diseases) group Diffuse Photoreceptor Dystrophies group. Please comment.

– What about cases where the phenotype changes or does not fit any of the proposed groups? Some PRPH2-associated macular dystrophies for instance. Shouldn’t there be a group for cases that cannot be classified anywhere else?

Supplementary table 1

– Is the term “regional retinitis pigmentosa” a synonym of “sector retinitis pigmentosa”? If yes, than the term sector retinitis pigmentosa should be used instead, as this is the most common term

References

– Duplicate references! Ref 8 and ref 21 are the same

6. PLOS authors have the option to publish the peer review history of their article (what does this mean?). If published, this will include your full peer review and any attached files.

Reviewer #1: No

---

## [Author Response · Author response to Decision Letter 0]

13 Oct 2022

PONE-D-22-12220

A Health Terminological System for Inherited Retinal Diseases: Content Coverage Evaluation and a Proposed Novel Classification

PLOS ONE

Dear Dr. Sheikhtaheri,

Thank you for submitting your manuscript to PLOS ONE. After careful consideration, we feel that it has merit but does not fully meet PLOS ONE’s publication criteria as it currently stands. Therefore, we invite you to submit a revised version of the manuscript that addresses the points raised during the review process. I am sorry that it has taken a long time to present you with a review, but we were trying to find a second expert to review your submission. In the interests of time, I decided to ask for revision based on the comments of one expert whose comments are valuable.

We look forward to receiving your revised manuscript.

Kind regards,

Alfred S Lewin, Ph.D.

Section Editor

PLOS ONE

Journal Requirements:

Editor comment 

Please modify your comments about the novelty of your work, because there are already several classification systems for IRDs, though none is widely accepted. Please explain how genetic data informed your classification scheme. The reviewer indicates that including genotypic information is important.

Answer: The IRD hierarchy has been revised based on both genetic and anatomical entities. Also, all required modifications have been performed throughout the text. Although there are several classification systems for rare eye diseases, no certain hierarchy has been developed specifically for IRD concepts. The current IRD classification has been developed based on the textbooks review, and the related standard terminologies such as Unified Medical Language System (UMLS), Online Mendelian Inheritance in Man (OMIM), International Classification of Diseases (ICD-11), Systematized Nomenclature of Medicine (SNOMED-CT) and Orphanet Rare Disease Ontology (ORDO) which is more comprehensive compared with other classifications. 

Reviewers' comments:

Reviewer's Responses to Questions

Comments to the Author

1. Is the manuscript technically sound, and do the data support the conclusions?

Reviewer #1: Partly

Answer: The conclusion part has been revised to be completely in line with the study findings. 

2. Has the statistical analysis been performed appropriately and rigorously?

Reviewer #1: N/A

3. Have the authors made all data underlying the findings in their manuscript fully available?

Reviewer #1: Yes

4. Is the manuscript presented in an intelligible fashion and written in standard English?

Reviewer #1: Yes

5. Review Comments to the Author

Reviewer #1: The authors present a novel IRD classification. The methodology used to achieve this new classification is well defined and all data was made available. Please find my queries below:

Abstract

– Purpose: Please use “To present a novel…” instead of “To represent a novel…”

Answer: It has been done. 

Introduction

– The authors write: “to the best of our knowledge there is no dedicated ontology or classification for IRD concepts”. This is not true. There are several proposed classifications. However, I would say there is not one widely accepted classification. Also, please check PMID: 30626441 (ref 8 in your references list). The authors present a summary of the modifications made to HPO and ORDO as part of the study (Table S1), and list the disease groups and specific disorders introduced to ORDO as a result of the ERN-EYE Ontology meeting (Mont Sainte-Odile, France, 10/2017). The ERN-EYE is an European Reference Network covering all rare eye disorders and not only IRDs. The group has worked hand in hand with ORPHANET to improve IRD and other rare eye disease nomenclature in ORDO.

Answer: Although there are several classification systems for rare eye diseases, no certain hierarchy has been developed specifically for IRD concepts. The current IRD classification has been developed based on the text books review, and the related standard terminologies such as Unified Medical Language System (UMLS), Online Mendelian Inheritance in Man (OMIM), International Classification of Diseases (ICD-11), Systematized Nomenclature of Medicine (SNOMED-CT) and Orphanet Rare Disease Ontology (ORDO) which is more comprehensive compared with other classifications.

– Please replace “…to validate the value of this novel…” for “…to validate this novel…”

Answer: It has been done. 

Methods

– Step I: the authors write “This proposed classification was structured based on the phenotype characteristics and the genetic concepts were included to the related IRD diagnosis.” Please elaborate on the genetic concepts. I cannot find in the text, tables, figures or supplemental material any reference to the genetic cause of the proposed diagnoses. In fact, I would say the major issue with the presented classification is that it lacks genetic information. Please elaborate on this as it is a fundamental part of any new IRD classification. Several genes are able to cause different phenotypes and gene-based classifications are getting more popular. Developing a strong phenotypical classification is important but it must incorporate genetic data, ie, genes associated with each phenotype and must allow seeing things the other way around, ie which phenotypes are associated with each gene.

Answer: The IRD hierarchy has been revised based on both genetic and anatomical entities. Also, all required modifications have been performed throughout the text.

– Another issue is that some diagnoses fit more than one category. An example would be central areolar choroidal dystrophy (CACD) which fits the macular dystrophies group and choroidal dystrophies group. The same is true for syndromic forms of retinitis pigmentosa (eg Usher, BBS) that fit the Retinal dystrophies associated with systemic disease (syndromic diseases) group and the Diffuse Photoreceptor Dystrophies group; or syndromic forms of cone-rod dystrophy (eg. Cone-rod dystrophy and hearing loss, caused by CEP250 mutations) that fit the Retinal dystrophies associated with systemic disease (syndromic diseases) group Diffuse Photoreceptor Dystrophies group. Please comment.

Answer: Thank you for raising this concern, which is a potential issue in any classification system. In our proposed classification system, each IRD disease is mainly grouped based on the main origin of its pathological problem considering both anatomical and functional concepts. Although, many IRDs gradually progress and after a while will involve other adjacent structures. For instance, as you indicated CACD is mostly a choroidal disease that gradually extends to involve the macular region or farther parts of the retina. Therefore, in our classification system, we grouped this diagnosis under choroidal dystrophies. In the same way, we classified the “diffuse photoreceptor dystrophies” as a group containing the IRDs that exclusively involve the retina with no other organ involvement, besides there is another group of “systemic diseases associated with photoreceptor dystrophies” to embrace retinal dystrophies that have an associated systemic or organ condition such as BBS and Usher syndrome or cone-rod dystrophy and hearing loss. Please refer to supplementary table 1. 

– What about cases where the phenotype changes or does not fit any of the proposed groups? Some PRPH2-associated macular dystrophies for instance. Shouldn’t there be a group for cases that cannot be classified anywhere else?

Answer: You have pointed out a great example. We reported a complex family affected with different clinical features of inherited retinal dystrophy, including fundus flavimaculatus as an early clinical feature progressing to extensive chorioretinal atrophy involving the macula and mid-periphery of the fundus in one parent and central areolar chorioretinal dystrophy (CACD) as the most probable clinical diagnosis in another parent. Macular pattern dystrophy for one of their daughters and a Leber congenital amaurosis (LCA) like phenotype for the daughter with an early onset retinal dystrophy (EORD) phenotype. (Daftarian N, et al. Ophthalmic Genetics, 40:5, 436-442). Therefore, in our proposed classification, the disease categorization was organized considering both anatomical and functional IRD entities in seven groups, instead of being exclusively based on the genetic, anatomic or functional diagnosis. Supplementary table 1

– Is the term “regional retinitis pigmentosa” a synonym of “sector retinitis pigmentosa”? If yes, then the term sector retinitis pigmentosa should be used instead, as this is the most common term.

Answer: Thank you for alluding to another precise point. We replaced it with the more familiar term “sector retinitis pigmentosa” and then will use the regional retinitis pigmentosa as the alternative one. 

References

– Duplicate references! Ref 8 and ref 21 are the same.

Answer: Many thanks, the list of references has been revised. 

6. PLOS authors have the option to publish the peer review history of their article. If published, this will include your full peer review and any attached files.

Do you want your identity to be public for this peer review? For information about this choice, including consent withdrawal, please see our Privacy Policy.

Reviewer #1: No

---

## [Decision Letter · Decision Letter 1]

1 Nov 2022

PONE-D-22-12220R1A Health Terminological System for Inherited Retinal Diseases: Content Coverage Evaluation and a Proposed ClassificationPLOS ONE

Dear Dr. sheikhtaheri,

Thank you for submitting your manuscript to PLOS ONE. After careful consideration, we feel that it has merit but does not fully meet PLOS ONE’s publication criteria as it currently stands. Therefore, we invite you to submit a revised version of the manuscript that addresses the points raised during the review process.

Please provide a table in which the genes associated with each proposed diagnosis are shown. Your final classification system should contain both anatomical and genetic classifications.

We look forward to receiving your revised manuscript.

Kind regards,

Alfred S Lewin, Ph.D.

Section Editor

PLOS ONE

Journal Requirements:

Reviewers' comments:

Reviewer's Responses to Questions

**Comments to the Author**

1. If the authors have adequately addressed your comments raised in a previous round of review and you feel that this manuscript is now acceptable for publication, you may indicate that here to bypass the “Comments to the Author” section, enter your conflict of interest statement in the “Confidential to Editor” section, and submit your "Accept" recommendation.

Reviewer #1: (No Response)

2. Is the manuscript technically sound, and do the data support the conclusions?

Reviewer #1: Partly

3. Has the statistical analysis been performed appropriately and rigorously? 

Reviewer #1: N/A

4. Have the authors made all data underlying the findings in their manuscript fully available?

Reviewer #1: Yes

5. Is the manuscript presented in an intelligible fashion and written in standard English?

Reviewer #1: Yes

6. Review Comments to the Author

Reviewer #1: Thank you for your revision. You have addressed most of my concerns. However, one major issue remains:

I still cannot find any table or supplemental table where the genes associated with each proposed diagnosis are shown. The link you provide with your final classification (https://bioportal.bioontology.org/ontologies/IRD1) separates the genetic classification (genetic_variantion) from the anatomical classification (Inherited_retinal_Dystrophy). It is my understanding that a novel classification should merge botth.

Let’s see a practical example: retinitis pigmentosa associated with biallelic EYS variants:

According to your proposed classification system, this would be:

IRD_1_1 but there are several causative genes so we should be able to select the causative gene (IRD_1_1#21 according to your genetic variation list) from the first classification.

Please provide a merged classification.

Minor issues

- Why do you separate AR and AD pericentral pigmentary retinopathy? I understand the same phenotype may be caused by AR or AD genes but if you are separating it here according to the inheritance pattern, then you should be doing the same for Retinitis Pigmentosa (AR, AD, XL), Cone-rod dystrophy (AR, AD, XL), etc

- Wouldn’t it be easier to join Choroidal dystrophies (IRD 3) with Chorioretinal dystrophies (IRD 7)?

- Regarding the "Supporting Information – Final Classification and Coverage Version 12.18.2021", the authors did not change the term “regional” to “sector” retinitis pigmentosa

7. PLOS authors have the option to publish the peer review history of their article (what does this mean?). If published, this will include your full peer review and any attached files.

Reviewer #1: No

---

## [Author Response · Author response to Decision Letter 1]

1 Feb 2023

PONE-D-22-12220R1

A Health Terminological System for Inherited Retinal Diseases: Content Coverage Evaluation and a Proposed Classification

PLOS ONE

Editor comment

Please provide a table in which the genes associated with each proposed diagnosis are shown. Your final classification system should contain both anatomical and genetic classifications.

Thank you for the comment. We added genes information. 

 

Reviewers' comments:

Reviewer's Responses to Questions

Comments to the Author

1. If the authors have adequately addressed your comments raised in a previous round of review and you feel that this manuscript is now acceptable for publication, you may indicate that here to bypass the “Comments to the Author” section, enter your conflict of interest statement in the “Confidential to Editor” section, and submit your "Accept" recommendation.

Reviewer #1: (No Response)

2. Is the manuscript technically sound, and do the data support the conclusions?

Reviewer #1: Partly

Answer: The conclusion part has been revised to be more corresponding with our findings. 

3. Has the statistical analysis been performed appropriately and rigorously?

Reviewer #1: N/A

4. Have the authors made all data underlying the findings in their manuscript fully available?

Reviewer #1: Yes

5. Is the manuscript presented in an intelligible fashion and written in standard English?

Reviewer #1: Yes

6. Review Comments to the Author

Reviewer #1: Thank you for your revision. You have addressed most of my concerns. However, one major issue remains:

I still cannot find any table or supplemental table where the genes associated with each proposed diagnosis are shown. The link you provide with your final classification (https://bioportal.bioontology.org/ontologies/IRD1) separates the genetic classification (genetic_variantion) from the anatomical classification (Inherited_retinal_Dystrophy). It is my understanding that a novel classification should merge both.

Let’s see a practical example: retinitis pigmentosa associated with biallelic EYS variants:

According to your proposed classification system, this would be:

IRD_1_1 but there are several causative genes so we should be able to select the causative gene (IRD_1_1#21 according to your genetic variation list) from the first classification.

Please provide a merged classification.

Answer: The merged IRDs hierarchy combining both anatomical and genetic codes providing based on five standard coding systems has been submitted to the website as well as the BioPortal website.

Minor issues

- Why do you separate AR and AD pericentral pigmentary retinopathy? I understand the same phenotype may be caused by AR or AD genes but if you are separating it here according to the inheritance pattern, then you should be doing the same for Retinitis Pigmentosa (AR, AD, XL), Cone-rod dystrophy (AR, AD, XL), etc.

Answer: The title of each IRD entity either includes the inheritance pattern or not has been selected based on the standard terminologies without any updates and no suggested terms have been inserted into the heicharcy. 

- Wouldn’t it be easier to join Choroidal dystrophies (IRD 3) with Chorioretinal dystrophies (IRD 7)?

Answer: These two categories have been merged into the one group labeled as "Choroidal Dystrophy".

- Regarding the "Supporting Information – Final Classification and Coverage Version 12.18.2021", the authors did not change the term “regional” to “sector” retinitis pigmentosa.

Answer: It has been changed as you recommended. 

7. PLOS authors have the option to publish the peer review history of their article (what does this mean?). If published, this will include your full peer review and any attached files.

Do you want your identity to be public for this peer review? For information about this choice, including consent withdrawal, please see our Privacy Policy.

Reviewer #1: No

---

## [Editor Report · Decision Letter 2]

3 Feb 2023

A Health Terminological System for Inherited Retinal Diseases: Content Coverage Evaluation and a Proposed Classification

PONE-D-22-12220R2

Dear Dr. sheikhtaheri,

We’re pleased to inform you that your manuscript has been judged scientifically suitable for publication and will be formally accepted for publication once it meets all outstanding technical requirements.

Kind regards,

Alfred S Lewin, Ph.D.

Section Editor

PLOS ONE
---

## [Editor Report · Acceptance letter]

7 Feb 2023

PONE-D-22-12220R2 

A Health Terminological System for Inherited Retinal Diseases: Content Coverage Evaluation and a Proposed Classification 

Dear Dr. Sheikhtaheri:

I'm pleased to inform you that your manuscript has been deemed suitable for publication in PLOS ONE. Congratulations! Your manuscript is now with our production department. 

Kind regards, 

on behalf of

Dr. Alfred S Lewin 

Section Editor

PLOS ONE